# Recent Advancement in mRNA Vaccine Development and Applications

**DOI:** 10.3390/pharmaceutics15071972

**Published:** 2023-07-18

**Authors:** Nojoud Al Fayez, Majed S. Nassar, Abdullah A. Alshehri, Meshal K. Alnefaie, Fahad A. Almughem, Bayan Y. Alshehri, Abdullah O. Alawad, Essam A. Tawfik

**Affiliations:** 1Advanced Diagnostics and Therapeutics Institute, Health Sector, King Abdulaziz City for Science and Technology (KACST), Riyadh 11442, Saudi Arabia; nalfayez@kacst.edu.sa (N.A.F.);; 2Healthy Aging Research Institute, Health Sector, King Abdulaziz City for Science and Technology (KACST), Riyadh 11442, Saudi Arabia

**Keywords:** mRNA vaccine, delivery, lipid nanoparticles (LNPs), infectious diseases, cancer, immune response

## Abstract

Messenger RNA (mRNA) vaccine development for preventive and therapeutic applications has evolved rapidly over the last decade. The mRVNA vaccine has proven therapeutic efficacy in various applications, including infectious disease, immunotherapy, genetic disorders, regenerative medicine, and cancer. Many mRNA vaccines have made it to clinical trials, and a couple have obtained FDA approval. This emerging therapeutic approach has several advantages over conventional methods: safety; efficacy; adaptability; bulk production; and cost-effectiveness. However, it is worth mentioning that the delivery to the target site and in vivo degradation and thermal stability are boundaries that can alter their efficacy and outcomes. In this review, we shed light on different types of mRNA vaccines, their mode of action, and the process to optimize their development and overcome their limitations. We also have explored various delivery systems focusing on the nanoparticle-mediated delivery of the mRNA vaccine. Generally, the delivery system plays a vital role in enhancing mRNA vaccine stability, biocompatibility, and homing to the desired cells and tissues. In addition to their function as a delivery vehicle, they serve as a compartment that shields and protects the mRNA molecules against physical, chemical, and biological activities that can alter their efficiency. Finally, we focused on the future considerations that should be attained for safer and more efficient mRNA application underlining the advantages and disadvantages of the current mRNA vaccines.

## 1. Introduction

Vaccination has played a pivotal role in combating infectious diseases and has contributed significantly to public health worldwide [1]. Traditional vaccine platforms primarily utilize inactivated or attenuated pathogens, protein subunits, or viral vectors, which have played a significant role in the disease prevention [2]. However, these conventional vaccine platforms often require complex manufacturing processes and may have limitations in terms of scalability, speed of development, and adaptability to emerging pathogens [3,4].

In recent years, mRNA (i.e., messenger RNA) vaccines have emerged as a groundbreaking technology, offering a versatile platform for rapid and targeted vaccine development against infectious diseases, including viral outbreaks and emerging pathogens [5,6]. The mRNA vaccines represent a new class of vaccines that harness the body’s cellular machinery to produce specific antigens, which will trigger the immune responses against the produced antigens [7]. This innovative approach offers numerous advantages over traditional vaccine platforms. It has garnered significant attention, especially in the context of recent successes, such as the development and deployment of mRNA-based vaccines against the SARS-CoV-2 virus [8,9,10]. One of the key advantages of mRNA vaccines is their rapid development and production capabilities. Unlike traditional vaccines, which often require time-consuming and costly manufacturing processes, mRNA vaccines can be designed and synthesized in weeks, offering unprecedented flexibility and speed in response to emerging infectious diseases or evolving variants [11]. Additionally, the modular nature of mRNA vaccines enables easy adaptation to different pathogens by simply altering the mRNA sequence encoding the desired antigen [12,13,14]. However, the efficient delivery of mRNA molecules into target cells poses a significant challenge, as they are susceptible to degradation, have limited stability, and face barriers in reaching the desired sites of action [15,16].

Nanoparticles have emerged as promising tools in mRNA vaccine delivery, offering numerous advantages to overcome the inherent limitations of naked mRNA molecules [17,18]. These nanoscale carriers provide protective and delivery capabilities, facilitating efficient cellular uptake, protection from enzymatic degradation, and controlled release of mRNA payloads. Additionally, nanoparticles can be engineered to improve stability, prolong circulation time and facilitate their targeted delivery to immune cells or specific tissues, ultimately leading to enhanced immunogenicity and efficacy of mRNA vaccines [19,20].

Lipid-based nanoparticles (LNPs) have emerged as the leading class of nanoparticles for mRNA vaccine delivery. LNPs can encapsulate and protect mRNA within their hydrophobic core, enabling efficient cellular uptake and intracellular release [21,22]. This approach has been successfully utilized in developing mRNA-based vaccines against various infectious diseases, including COVID-19 [6,23]. LNPs have shown excellent safety profiles, high transfection efficiency, and the ability to induce robust immune responses, positioning them as promising tools for effective vaccination strategies.

Beyond LNPs, other types of nanoparticles, such as polymeric nanoparticles, peptides, and protamine-based delivery systems, have also been explored for mRNA vaccine delivery. Each nanoparticle class offers distinct advantages, such as tunable size, surface modifications, and targeted delivery capabilities. Polymeric nanoparticles, including synthetic polymers and natural polymers, provide versatility in design and can protect the naked mRNA from degradation [24]. At the same time, self-assembling peptides and protamine nanocomplexes offer unique properties in the cell-penetrating [25,26,27,28].

The importance and relevance of nanoparticle mRNA vaccines extend beyond their application in infectious disease prevention. The mRNA technology holds promise for addressing other healthcare challenges, including cancer immunotherapy, personalized medicine, and therapeutic interventions for genetic disorders [29,30,31,32]. The ability to rapidly design and manufacture mRNA vaccines using nanoparticles has demonstrated its potential for agility in responding to emerging infectious diseases, making it a valuable asset in the pandemic preparedness [33].

In this review, we aim to provide a comprehensive overview of the role of nanoparticles in mRNA vaccine delivery. We will address the definition of mRNA vaccines, mechanism of action, delivery systems, manufacturing process, and clinical applications. By elucidating the importance and relevance of mRNA vaccines using nanoparticles, this review will also aim to shed light on the advancements in vaccine development and the potential for transformative approaches in infectious disease prevention and other therapeutic interventions.

## 2. Development of mRNA Vaccines

In this section, the various types of mRNA vaccines will be discussed, and their mechanisms of action and the different approaches that can be used to enhance the safety and efficacy of the mRNA vaccines will be illustrated.

### 2.1. Types of mRNA Vaccines

Enhancing immunity by genetic vaccination can be superior to traditional immunization, such as recombinant proteins and live or inactivated viruses [34]. For example, a DNA vaccine can be ideal because it can stimulate both the humoral and cellular immunities [34]. Furthermore, the large-scale manufacturing of genetic vaccines might be more tolerated than traditional vaccines [35]. There are three types of mRNA vaccines: conventional mRNA, self-amplifying mRNA, and trans-amplifying mRNA [35]. The conventional mRNA contains the untranslated regions (5′UTR, 3′UTR) and the coding region of mRNA that can be transcribed to one copy of immunogenic protein; therefore, it is called the non-amplifying or non-replicating mRNA [35]. The immune response is directly proportional to the number of transcribed mRNAs, which may require a high dose of mRNA due to its unamplified behavior that could cause repeated administration of mRNA [35]. The self-amplifying mRNA vaccine is a genetically modified mRNA by adding engineered replicons originating from the self-replicating RNA viruses [36,37]. Both self-amplifying mRNA and trans-amplifying mRNA have 5′ and 3′ conserved sequence elements (CSE) [35]. It is hypothesized that CSEs can regulate viral RNA synthesis in addition to their ability to attach to viral or cellular proteins [38]. The main difference between self-amplifying mRNA and trans-amplifying mRNA is that the self-amplifying mRNA has non-structural proteins 1-4 (nsP 1-4) sequence, whereas the trans-amplifying mRNA requires two RNA genes to be co-delivered, the mRNA without nsP 1-4 and the mRNA that encodes nsP1-4 genes [39].

### 2.2. Mechanism of Action of mRNA Vaccines

Naturally, mRNA in a cell is a transcribed gene from the genomic DNA in the nucleus, and it can be translated by the ribosome in the cytoplasm to form the required protein [40]. Initially, the mRNA is transcribed from genomic DNA as a primary mRNA with the aid of RNA polymerase. The primary mRNA contains exon and intron. Thereafter, further modification of the primary mRNA occurs to be converted to mature mRNA, such as splicing of primary mRNA to remove the non-coding introns, capping the 5′UTR, and poly-A sequence addition at the 3′ terminal [41].

In mRNA vaccine technology, the mRNA is synthesized outside the human body and is subjected to be injected and crossed the cell membrane for translation in the cytoplasm. After the mRNA is transferred from the nucleus, in the case of naturally synthesized mRNA, or from the cell membrane, in the case of mRNA vaccine to the cytoplasm, the mRNA is translated to the required protein by the ribosome. In the translation process, the poly-A tail binds to the poly-A-binding protein (PABP), and the eIFs bind to the 5′UTR cap to initiate the translation. The ribosome then translates each codon in the translated region of the mRNA, which consists of three nucleotides, to amino acid [42]. From the mechanism of action of the mRNA translation, it is important to note that the contents of the synthesized mRNA vaccine that contains 5′ cap, 5′UTR, translated region, 3′UTR, and poly-A tail to be recognized by the ribosome for successful translation of mRNA to the required antigen [43].

The main advantage of using mRNA compared to the plasmid DNA (pDNA) vaccine is that the pDNA requires further steps by internalizing inside the cell nucleus and transcribing into the required mRNA successfully before the translation step to the protein of interest [44]. Several physiological barriers hinder the delivery of the synthesized RNA. One of the main barriers preventing RNA delivery is crossing the cell membrane and escaping from the endosomes [45]. Therefore, mRNA requires a delivery carrier which will be explained in detail in the next section (Section 3). In the conventional mRNA, the open reading frames (ORFs) encode the antigen that can be translated to the antigen of interest, such as a virus epitope, which causes an immunogenic reaction that will lead to the recognition of the actual pathogen when the vaccinated subject gets infected with it [40]. Figure 1 illustrates the mechanism of immune system activation in response to mRNA administration.

The self-amplifying mRNA contains the genetic sequences responsible for the mRNA replication before the translation step, which is the CSE and the nsP 1-4 [35]. The nsP1 is required for 5′ capping of the viral RNA, and nsP2 can unwind the RNA duplex during replication in addition to the helicase and protease activity [46]. Furthermore, nsP2 can cleave the nsP 1-4 polyprotein into its subunits (nsP1, nsP2, nsP3, and nsP4) [46]. The nsP3 facilitates the interaction between viral and host proteins [47], while nsP4 is an RNA-dependent RNA-polymerase (RdRp) that catalyzes the replication of RNA [39]. After the self-replication of the self-amplifying mRNA into several copies, these copies will be translated into antigens that will cause a more immunogenic response than the conventional mRNA [48,49].

### 2.3. Optimizing mRNA Vaccines

Different approaches can enhance the stability, safety, and efficiency of mRNA vaccines. Modifying the untranslated regions of the mRNA could play a vital role in increasing its stability and, thus, its efficacy [5]. Even though the 5′ and 3′ UTRs do not have translated sequences during the translation process of mRNA, they can play a vital role in improving the stability of mRNA and protein expression from the mRNA [50,51]. Both the 5′ cap in the 5′UTR and the poly(A) end in the 3′UTR of the mRNA can regulate the translational efficiency of the mRNA [52]. For instance, it was reported that anti-reverse cap analogs were modified in the 5′UTR, which will enhance the translation of mRNA in comparison to the unmodified mRNA. This translation enhancement might be caused by increasing the binding affinity of the modified cap to eukaryotic translation initiation factors (eIFs) in the ribosomes [53]. Additionally, it was shown that changing the mRNA cap into phosphorothioate may improve the stability and the expression of the mRNA vaccine when it is transfected in immature dendritic cells [54]. Adding nucleotides to the mRNA can alter the stability and translation capacity. It was reported that inserting pseudouridine into the mRNA in mammalian cells increased the yield of the translated protein and decreased the unwanted immunogenic reaction compared to the unmodified mRNA. This could enhance both safety and efficacy by decreasing the immunogenic reaction and increasing the expression of the protein, respectively [55]. Furthermore, it was demonstrated that incorporating the modified nucleosides into the mRNA could reduce or eliminate the Toll-like receptors (TLRs) activation, which would enhance the safety of the mRNA vaccine by modification of its nucleosides [7].

## 3. The mRNA Vaccine Delivery Systems

This section will address four different types of materials, i.e., polymers, peptides, protamine, and lipids, and their respective delivery systems as carriers for the naked mRNA to enhance its stability, efficacy, and safety upon administration.

### 3.1. Polymeric Nanoparticles

Polyplex was initially used with polyethylenimine (PEI) polymer to deliver nucleic acids [56]. However, its use was limited due to its high surface charge density and toxicity profile. Researchers have explored alternative methods, such as adding polyethylene glycol (PEG) to the low molecular weight PEI and binding it to the cyclodextrin [57]. The conjugation with cyclodextrin was demonstrated to be an effective and safer approach for mRNA delivery, and it can be administered through different routes, which may lead to the production of distinct antibodies. New biodegradable polymers, such as poly(amino esters), have exhibited a promising future for delivering mRNA to the lungs [58]. Scientists have developed poly(-amino esters) libraries for structure–function studies, which can be produced using the Michael reaction [59]. These biodegradable polymers can easily modify the backbone and peripheral groups [60]. Hyper-branched dendritic spherical dendrimers with high amine density efficiently form mRNA complexes. However, their excessive charge densities harmed cells and caused serum clumping, but these challenges could be tackled by adding PEG to their cores or disulfide linkages [61].

The mRNA delivery uses pH-responsive polymers in another approach to utilize polymers, which can release the mRNA into the cells upon arrival to the cytoplasm by self-degrading due to the cytoplasmic pH [25]. Amphipathic polymers containing cationic or amphipathic amine groups can also carry mRNA into the cells by electrostatically binding to the mRNA, creating nano-complexes. Generally, high molecular weight polymers, such as chitosan and PEI, need modification to improve the transfection efficacy and durability [62]. At the same time, the mRNA affinity can be influenced by the polymer’s length, charge density, and mixture concentration [63,64,65,66].

### 3.2. Peptides and Proteins Nanoparticles

Peptide/protein-based biological nanomaterials have been employed extensively in drug delivery, disease diagnosis, and vaccine development due to their excellent biocompatibility and relative accessibility [67,68,69]. The protein nanoparticles typically have 20 to 200 nm dimensions, ideal for effective lymph node targeting. A reliable and secure method of administering vaccines is using self-assembled protein carriers in vaccine delivery systems. For instance, Wang et al. have reported that a hepatitis B virus (HBV) vaccination with both therapeutic and preventative properties was achieved by the conjugation of Pre-S1 HBV surface antigen and self-assembled ferritin nanoparticles (NP-PreS1) [70].

Amphipathic peptides can help deliver mRNA into cells owing to their cationic or amphipathic amine groups, such as arginine, which can electrostatically bind to the mRNA, creating nano-complexes. The endosome’s pH level can alter the fusion cell-penetrating peptide (CPP) structure with repeated arginine-alanine-leucine-alanine (RALA) motifs. It can, thereby, facilitate the creation of pores between membranes and endosomes, enabling the delivery of mRNA into cells [71]. Additionally, RALA can activate T-cell-mediated immunity and transport mRNA to dendritic cells [25]. A recent study of amphipathic CPP/mRNA complexes, with a size of less than 200 nm, has shown a significant cellular uptake and protein expression in mice colon cancer cells [61].

### 3.3. Protamine Nanoparticles

Protamine is a cationic protein that is composed mainly of positively charged amino acids [27]. This net positive charge makes protamine able to complex nucleic acids, such as RNAs, to improve the stability of biological systems by protecting them from enzymatic degradation by nuclease and enhancing their delivery to targeted tissue [72]. Due to the versatile features and clinical safety of protamine compound, it has multiple applications in biomedical research as a drug delivery nanocarrier [73]. The most important feature of the protamine delivery system is the cationic properties due to an arginine-rich sequence, which enhances the complexation of protamine with any negatively charged molecules, such as mRNA, which can be exploited in the designing of mRNA-based vaccines [28].

Protamine-mRNA-utilizing galactosidase enzyme was injected into a glioblastoma tumor to assess the viability of the mRNA protamine complex [74,75]. The studies showed that protecting the mRNA with protamine resulted in a significant decrease in the tumor’s growth and eventual disappearance, as well as a memory antitumor immune response developing after intra-tumoral injection of the stabilized mRNA. Furthermore, protamine was used with CureVac’s self-adjuvanted RNActive^®^ delivery technology to complex mRNA, which led to the development of rabies and influenza A vaccines [76,77]. It should be noted that protamine has been used to improve the transfection efficiency of complexed nucleic acid with no detectable cytotoxicity of concentration up to 10 mM in comparison to other commonly used transfection reagents (i.e., Polyethylenimine (PEI) polymer) which reported to induce in vitro toxic effect at dose above 5 mM [27]. Recently, the exploitation of protamine in the delivery of mRNA has been increased to prepare well-established vaccines, and more research is expected. 

### 3.4. Lipid Nanoparticles (LNPs)

The LNPs are smart, nanoscale lipid-based carriers for mRNA transport into the cytosol. These particulate nanocarriers can effectively deliver mRNA intracellularly by fusing with the lipid bilayer of the early endosomes, providing the mRNA into the cytoplasm, and safeguarding the mRNA against RNase during systemic circulation [78]. The LNPs have been employed in developing SARS-CoV-2 mRNA vaccines that have received clinical approval. For mRNA administration, LNPs provide substantial advantages, such as simplicity of formulation, modularity, biocompatibility, and high mRNA payload capacity. The LNPs typically consist of three main components: an ionizable lipid (40–50%); cholesterol (38–45%); and a helper phospholipid (10–12%); in some cases, a fourth component is added, such as PEGylated lipid (1–2%) [79], as illustrated in Figure 2. These components function together to enclose and safeguard the naked mRNA [48]. The first lipids that carried an RNA were the cationic DOTMA (1,2-Di-O-octadecenyl-3-trimethylammonium propane) and their synthetic equivalent DOTAP (1,2-Dioleoyl-3-trimethylammonium-propane) in 1989 [80]. Since then, numerous additional cationic lipids, such as the widely used Lipofectamine, have been employed for RNA delivery [81].

Cationic lipid amphiphiles have been created and evaluated as carriers of nucleic acids. The cationic lipids have a similar molecular structure to natural lipids, except for holding a cationic head group instead of the natural lipids’ zwitterionic or anionic head group. They are formed by a linker between the polar group and the hydrophobic moiety, a positively charged polar head group. In contrast, the hydrophobic moiety contains two alkyl chains or a cholesterol moiety. The cationic lipids are preferable for drug delivery applications due to their positively charged surfaces that bind with the negatively charged cell surface, facilitating the permeation of the LNPs into the cells [82].

The LNPs were utilized for delivering mRNA vaccines and drugs against COVID-19, influenza, and tuberculosis [5,83]. When ionizable lipids are introduced into the bloodstream at physiological pH levels, they have a neutral charge, which enhances their safety and extends their circulation time compared to other lipids [18]. These ionizable lipids are combined with mRNA in an acidic solution, forming nanoparticles in which the positively charged lipids attract the RNA. Moreover, due to the acidic environment of the endosomes, they readily fuse with the endosomal membrane, releasing the mRNA into the cytoplasm [84]. The first ionizable lipids utilized for RNA transport were DODAP (1,2-Dioleoyl-3-dimethylammonium-propane) and DODMA (1,2-Dioleyloxy-3-dimethylaminopropane) [85]. Another ionizable lipid used for mRNA delivery is DLin-MC3-DMA owing to its high efficacy and safety profiles when encapsulated siRNA [86,87,88,89].

The science of mRNA delivery may transform because of the revolutionary technology known as microfluidics. Microfluidic devices are anticipated to create LNPs for medicinal and vaccination purposes, increasing their frequency as technology advances. The loading of mRNA into LNPs is accomplished through microfluidics, which involves precise control of the mixing of mRNA and LNP components and results in proper containment of the mRNA within the LNPs. Additionally, microfluidic enables the creation of LNPs with a homogenous size distribution, which is crucial for their effective delivery of mRNA into cells [90,91]. The typical procedure for preparing mRNA-loaded LNPs involves dissolving mRNA in a low pH aqueous buffer (pH~4.0) and microfluidically mixing it with an ethanolic solution of hydrophobic lipids to form stable LNPs with low polydispersity index. These LNPs initially contain 25–50% ethanol and have a low pH; thus, they require further dialysis and buffer exchange using tangential flow filtration (TFF) to obtain the final bulk product [18,78]. This product can be sterilized by filtering through a 0.2 μm sterile grade filter and then filled into aseptic containers. The finished drug product can be in either a lyophilized or direct fill–finish format [92].

## 4. The mRNA Vaccine Applications

The mRNA vaccines have been widely studied and developed for different diseases, including infectious diseases and personalized cancer vaccines. The potential applications of mRNA technology are anticipated to increase with ongoing developments in mRNA alterations and delivery techniques.

### 4.1. The mRNA Vaccines against Infectious Diseases

The COVID-19 pandemic has accelerated research efforts and prompted significant interest in the possibility of mRNA infectious disease vaccines, which have long been under development. The most prominent and extensively utilized examples are the Pfizer-BioNTech and Moderna COVID-19 vaccines that encode the spike protein of SARS-CoV-2 to stimulate an immune response against the virus [93]. The mRNA vaccine encodes the antigen of the spike protein or the spike protein’s receptor-binding domain, therefore protecting the infected subject from this life-threatening infection. In late 2020, the FDA approved the first two mRNA vaccines for SARS-CoV-2, the Pfizer–BioNTech vaccine (BNT162b2) and the Moderna vaccine (mRNA-1273). Both vaccines were formulated as LNPs encapsulating the mRNA using ionizable lipids. Two doses of BNT162b2 proved to be 87% effective at preventing hospitalizations, 94% effective at preventing symptomatic COVID-19, and 95% overall effective at preventing COVID-19 [9,94]. Primary and booster injections of both BNT162b2 and mRNA-1273 were reported to be 90% effective against SARS-CoV-2 [95]. It should be noted that the Moderna vaccine showed good safety and effectiveness following its storage for a month at 4–8 °C, while the Pfizer–BioNTech vaccine needs −60 °C storage conditions. Several more mRNA vaccines for infectious diseases, such as influenza, Zika, Cytomegalovirus, and rabies, are being developed [96]. A recent study developed an LNP-based vaccine from the 20 lineages of influenza A and B viruses encoding hemagglutinin antigens to protect against homologous and heterologous variable viruses [97].

### 4.2. The mRNA Vaccines against Cancers

The target of mRNA cancer vaccines depends on the tumor antigens to stimulate cellular immunity. Two groups of tumor antigens can be distinguished, tumor-specific antigens (TSA), which are only expressed on tumor cells, and tumor-associated antigens (TAA), which refer to the antigen molecules found on normal or tumor cells [98]. The FixVac BNT111 vaccine, which BioNTech is developing, is one example of an mRNA cancer vaccine that targets four melanoma TAAs, including tyrosinase, melanoma antigen family A3 (MAGE A3), New York esophageal squamous cell carcinoma 1 (NY-ESO-1), and transmembrane phosphatase with tensin homology (TPTE) [99]. According to a preclinical study, BNT111 was shown to induce a potent antitumor immune response in a phase 1/2 trial, where 75% of the patients exhibited immune responses against at least one of the four TAAs following a series of eight injections, and CD8^+^ T cells were essential in the T-cell responses. Additional examples of mRNA vaccines include the BNT112 and BNT113 (i.e., individualized vaccines) that are tailored to the specific mutations presented in an individual patient’s tumor and encode five prostate cancer-specific antigens (PAP, PSA, and three undisclosed antigens) and HPV16-derived tumor antigens (HPV16 E6 and E7 oncoproteins), respectively. Moreover, a BNT122 is a combination vaccine that is designed to encode individual tumor mutations and a personalized set of pre-manufactured non-mutated shared tumor-associated antigens (BNT114), which is currently being evaluated in clinical trials for its safety, tolerability, and efficacy in treating various types of cancer [100,101]. The different vaccine candidates target multiple types of cancer, such as melanoma, breast cancer, pancreatic cancer, etc.

## 5. Clinical Trials of mRNA Vaccines

The mRNA vaccines can be implemented to control the spread of several life-threatening diseases, such as infectious diseases and cancer, and this type of vaccine can also apply to immunotherapies [102]. However, the field of mRNA vaccine remains mainly focused on infectious diseases, with around 70% among all ongoing mRNA vaccine trials. The number of mRNA vaccine-based clinical trials have been raised significantly in recent years following the COVID-19 pandemic and the success of the mRNA vaccine. The recent FDA approval of two rapid-response mRNA vaccines against COVID-19 is a remarkable achievement that facilitated the entry of more mRNA-based vaccines into clinical trials to be used for different medical applications [100]. The mRNA vaccines for infectious diseases are commonly administrated as a bolus injection, where they are translated into antigens after being taken up by immune and non-immune cells. In the clinical trials, many mRNA vaccine candidates have been submitted and investigated against infectious diseases and cancers, and the most popular examples are summarized in Table 1.

One of the infectious diseases that emerged as a global health crisis was the Zika viral infection in 2015. This infection caused mild to severe symptoms, such as fever and fatigue that could develop into fetal death during pregnancy [103]. It has been demonstrated that immunization against an antigen (i.e., prM-E protein) from any Zika virus strains could protect against all infections caused by the Zika virus since the disease is caused by a single serotype [104]. Encapsulating modified mRNA encoding Zika virus structural genes into lipid-based nanocarriers is a versatile platform to produce mRNA-based Zika vaccine [105]. The administration of prime-boost vaccine of modified mRNA encoding Zika prM-E genes that generated virus-like particles led to an increase in the levels of Zika virus neutralizing antibodies (nAbs) that protect immunocompromised and immunocompetent mice. Hence, the administrated mRNA Zika vaccine-induced protective antibody responses against the Zika virus [106]. It has been reported that a dose of 30 µg or 50 µg mRNA vaccine composed of LNPs loaded with prM-E mRNA generated neutralizing antibodies, which is 50–100 times higher than those induced by the conventional way of the inactivated virus [21]. Exploiting the mRNA platform as a Zika vaccine candidate showed potent neutralizing antibody responses to the Zika virus, providing a proof of concept for rapid protection against this infection [107]. Despite the effectiveness of this vaccine, it is poorly designed, in which it suffers from the activation of dengue viral infection via the production of cross-reactive antibodies [108]. The new Zika virus vaccine in a clinical trial has been developed by Moderna via the insertion of a mutated fusion loop epitope in the E protein into the modified prM-E mRNA (mRNA-1893). The result of mice immunization using this mRNA-based vaccine exhibited protection against the Zika virus with no production of dengue-enhancing antibodies [106].

One of the crucial advantages of the mRNA vaccine is the successful targeting of viral infection induced by different proteins simultaneously. For instance, the cytomegalovirus (CMV) vaccine (mRNA-1647) that Moderna has designed contains a cocktail of six different mRNA species, one encoding the CMV glycoprotein B (gB) and the other five species encoding five different proteins that form pentameric protein [109,110]. A new vaccine candidate (mRNA-1345) that encodes the perfusion F protein has been evaluated by Moderna to vaccinate children against the respiratory syncytial virus (RSV) [111]. This product has been formulated as LNPs and is currently between phases 2 and 3 of clinical trials [112].

An unmodified mRNA sequence encoding the full-length spike protein was also incorporated into LNPs to formulate CureVac’s vaccine candidate (CVnCoV) for COVID-19, which showed good stability in refrigerators for three months. Two doses of CVnCoV exhibited neutralizing antibodies similar to those recovering from COVID-19 [113]. Despite the good properties of this vaccine candidate, the clinical trial study showed that the effectiveness of CVnCoV against COVID-19 is around 47% only. Arcturus Therapeutics Company produced a self-amplifying mRNA vaccine candidate loaded into LNPs (ARCT-021), which encodes the full-length perfusion spike protein. A single dose of this vaccine candidate significantly protected injected mice from SARS-CoV-2 infection by elevating the IgG titers and high T cell response [114], resulting in full immunization following a single injection.

In addition to infectious diseases, cancer is one of the primary diseases that mRNA-based formulations could target and control. Several vaccine candidates were developed to treat cancer types, such as melanoma and non-small cell lung cancer. BNT122 is one of the products generated by BioNTech-Genentech and demonstrated a favorable safety profile. It contains around 20 specific-patient neo-epitopes administrated intravenously (I.V.) for treating skin cancer (i.e., melanoma) [115,116]. Another personalized mRNA-based cancer vaccine is mRNA-4157, which contains up to 34 neo-antigens encoded in a single mRNA strand. This vaccine was developed by collaborative work between Moderna and Merck, and the mRNA was incorporated into a lipid-based delivery vehicle administrated via intramuscular injection (I.M.) [117]. The personalized mRNA-based cancer vaccine (mRNA-4157) in combination with monoclonal antibodies chemotherapy pembrolizumab (Keytruda) enhanced recurrence-free survival (RFS) compared with pembrolizumab alone when used as an adjuvant treatment in patients with resected high-risk melanoma [118].

As explained in the previous section, BNT111 is a cancer vaccine candidate targeting the tumor-associated antigens TAAs usually expressed on the surface of tumor cells. This mRNA vaccine consists of a mix of four different melanoma-related antigens that could enhance the targeting efficiency. BNT111 is usually administrated via I.V. alone or in combination with cemiplimab, an anti-PD-1 monoclonal antibody being co-developed by Regeneron and Sanofi, to enhance the therapeutic efficacy against melanoma. The Phase 2 trial evaluated BNT111 in combination with cemiplimab against both agents as monotherapy in patients with anti-PD1-refractory/relapsed unresectable Stages III or IV melanoma. Survival can be as short as six months in the refractory or relapsed setting, depending on risk factors. Up to 50% of patients progress after treatment with checkpoint inhibitors [115]. Repeated doses of the BNT111 vaccine exhibited induction of immune response against the antigens via increased secretion of IFN-γ, IFN-α, IL-6, TNF- α, and other cytokines in patients’ plasma [115]. Furthermore, CV9202 is a vaccine candidate developed by CureVac composed of six different species of mRNA that encode six different antigens to target the non-small cell lung cancer [119].

The mRNA can also be applied as an immunotherapy, mainly against cancers. Several mRNA-based candidates were investigated in clinical trials, indicating significant potential for using this advanced technology in therapeutics. In Table 1, four products (BNT141, BNT311/GEN1046, BNT312/GEN1042, and BNT211) were investigated for efficient immunotherapy against cancers. BNT141 product developed by BioNTech reported being effective against different types of cancer, such as non-small-cell lung cancer, gastroesophageal junction adenocarcinoma, metastatic gastric cancer, oesophagal cancer, gallbladder cancer, pancreatic cancer, biliary tract cancer, colorectal cancer, and mucinous ovarian cancers, through targeting cells expressing CLDN18.2 [120]. The BNT311/GEN1046 and BNT312/GEN1042 are mRNA immunotherapies, which are investigational factors for bispecific antibody PD-L1 × 4-1BB and bispecific antibody CD40  ×  4-1BB, respectively. It was reported that they could stimulate the proliferation of T cells, secretion of cytokines, and antitumor activity. These two immunotherapy candidates demonstrated pharmacodynamic immune effects in the peripheral blood following their administration, suggesting the ability to be used to treat patients suffering from advanced refractory solid tumors [121,122]. The BNT211 is another immunotherapy candidate produced by BioNTech and can act by targeting the tumor-specific antigen Claudin-6 (CLDN6). The administration of this vaccine exhibited a high disease control rate of 86% and an overall response rate of 43% [123].

**Table 1 pharmaceutics-15-01972-t001:** mRNA vaccines under clinical trials (up to 1 June 2023).

**Name of Product**	**ClinicalTrials.Gov Number**	**Payload**	**Disease**	**Phase**	**Route of Administration**	**Trial Status**	**Sponsor**	**Reference**
**Infectious diseases**								
**mRNA-1893**	NCT04917861	Structural proteins of the Zika virus	Zika virus	2	I.M.	Active, recruiting	Moderna	[124]
**mRNA-1647**	NCT04232280	Six mRNA codings for pentamer viral antigen and gB protein of Cytomegalovirus	Cytomegalovirus infection	2	I.M.	Active, recruiting	Moderna	[125]
**mRNA-1345**	NCT05127434	The stabilized prefusion F protein	Respiratory syncytial virus	2–3	I.M.	Active, recruiting	Moderna	[126]
**CVnCOV**	NCT04652102	SARS-CoV-2	SARS-CoV-2	2–3	I.M.	Active, not recruiting	CureVac	[127]
**ARCT-021**	NCT04668339	SARS-CoV-2	SARS-CoV-2	2	I.M.	Active, not recruiting	Arcturus	[128]
**BNT162b2**	NCT04380701	SARS-CoV-2	SARS-CoV-2	1–2	I.M.	Active, recruiting	BioNTech–Pfizer	[129]
**mRNA-1273**	NCT04785144	Codes for the full-length prefusion stabilized S protein of the SARS-CoV-2 B.1.351 variant.	SARS-CoV-2 B.1.351 variant	2	I.M.	Active, not recruiting	Moderna	[130]
**Cancer**								
**BNT122**	NCT03815058	20 patient-specific neoantigens	Melanoma	2	I.V.	Active, recruiting	BioNTech– Genentech	[115]
**mRNA-4157**	NCT03897881	Encode up to 34 neoantigens	Melanoma	2	I.M.	Active, recruiting	Moderna–Merck	[51]
**BNT111**	NCT04526899	A mix of four melanoma-associated antigens	Melanoma	2	I.V.	Active, recruiting	BioNTech	[115]
**CV9202**	NCT03164772	NY-ESO-1, MAGE C1, MAGE C2, TPBG (5T4), survivin, MUC1	Non-small cell lung cancer	1–2	I.D.	Active, not recruiting	CureVac, Ludwig Institute	[131]
**BNT141**	NCT04683939	Secreted IgG antibody	Cancer	1–2	I.V.	Not yet recruiting	BioNTech	[100]
**BNT311/GEN1046**	NCT03917381	Bispecific antibody PD-L1 × 4-1BB	Cancer	1–2	I.V.	Active, recruiting	Genmab–BioNTech	[121]
**BNT312/GEN1042**	NCT04083599	Bispecific antibody CD40 × 4-1BB	Solid tumors	1–2	I.V.	Active, recruiting	Genmab–BioNTech	[100]
**BNT211**	NCT04503278	CAR T for CLDN6 + tumor + mRNA for CLDN6	CLDN6+ tumors	1–2	I.V.	Active, recruiting	BioNTech	[100]

Abbreviations: I.M., Intramuscular; I.V., Intravenous; I.D., Intradermal.

One of the main concerns regarding the administration of mRNA-based vaccines is the adverse events that might be induced following immunization (i.e., anaphylactic reaction). It has been reported that Pfizer–BioNTech and Moderna anti-COVID-19 mRNA vaccines increased the anaphylactic reaction by 2–4 times than usually seen by traditional vaccines [132,133]. The anaphylactic reaction could be promoted due to the antibodies produced against the components of mRNA delivery nanocarriers used or the mRNA molecule itself that could facilitate severe immune response [133]. Hence, the optimization of nanocarriers’ compositions and performing mRNA molecular modifications are crucial steps to avoid or minimize adverse reactions. The Centers for Disease Control and Prevention (CDC) suggested that people with a history of allergic reactions to the Pfizer–BioNTech or Moderna vaccines should avoid the administration of other mRNA-based vaccines [96].

In low-income countries, protection against infectious diseases is known as the most significant challenge that these countries face. The difficulty in providing certain storage conditions, such as −20 °C, is one of the main issues that should be addressed in the vaccines industry in general and mRNA vaccine manufacturing. Hence, this raises the need to develop a thermostable vaccine that can be stable at room temperature for an extended period to enhance global access to mRNA vaccines. Finally, it should be noted that several fatal infectious diseases worldwide have been eradicated in recent decades by vaccines. However, some anti-vaccination movements could also spread incorrect information and the re-emergence of life-threatening diseases. 

## 6. Future Applications of mRNA Vaccines

The successful development of COVID-19 mRNA vaccines has led to innovations in the future development of mRNA, which can produce next-generation mRNAs with more robust expression and excellent stability. These innovations can also increase the efficiency of mRNA delivery in preclinical studies, thereby accelerating their clinical translation.

### 6.1. Future Consideration in mRNA Vaccines Development

#### 6.1.1. Utilizing Different Types of mRNA

Self-amplifying mRNA includes a replicon based on Alphavirus that can amplify encoded protein expression, which requires significantly a lower dosage than most conventional mRNA applications [35]. Self-amplification mRNAs are more significant than traditional mRNAs due to incorporating additional replicon genes, and nucleoside modification cannot accommodate these modifications [35]. Several candidates are currently being evaluated in clinical trials [134]. SARS-CoV-2 vaccines based on self-amplifying mRNA have already been shown to induce high levels of neutralizing antibodies in animals. These vaccines can be administered in smaller doses (from 1 to 10 gm) than established mRNAs (from 30 to 100 gm) of SARS-CoV-2 vaccines [134].

Circular messenger RNAs (circ.mRNAs) are single-stranded RNAs structured as a circle and produced by the back-splicing process [135]. Recent research has shown that the protein-coding function of certain circ.mRNAs holds great promise for application in the protein translation [136] due to the absence of the terminus motifs required for the degradation process by exonuclease, circ.mRNAs confer more excellent stability than linear mRNAs [137,138]. Circular RNAs induce fewer unwanted immune responses than unmodified linear mRNAs and a circ.mRNA vaccine causes a more significant amount of neutralizing antibodies compared to a linear mRNA vaccine [139]. Companies are investigating additional variants of circ.mRNAs, including optimized internal ribosomal entry site (IRES) [140].

#### 6.1.2. Ideal Delivery System for mRNA

The LNPs are considered among the most prevalent system for mRNA delivery. Non-LNP systems, such as the retrovirus-like protein PEG10, have a potential for mRNA delivery owing to minimal immune stimulation and toxicity [141]. EntosPharma company created a fusogenic vehicle platform proteo-lipid applying neutral lipids with low-toxicity and exclusive fusion-associated small transmembrane (FAST) proteins to deliver mRNAs directly into the cytoplasm [142]. A redox-responsive coacervate and pH produced from a phase-separating peptide has accomplished direct delivery of mRNAs in the cytosol to cells and redox-activated mRNA release [143]. Innovations are continuing for the production of LNPs with various purposes that are more potent [144]. Heterocyclic lipids, unsaturated lipids, and alkyne lipids can improve the efficacy of LNPs. At the same time, their structural modifications with thiol groups or bisphosphate groups can target mRNAs to the mucus or bone [145,146]. An amphiphilic one-constituent ionizable dendrimer Janus has efficiently enabled the mRNAs delivery to various tissues, suggesting a potential de-complexification of the existing LNP four-constituent system [147].

Biocompatible vehicles are made of biological membranes that create a novel mRNA delivery platform. Different biological membrane carrier systems, comprising vesicles of the cell membrane, vesicles of the bacteria-based outer membrane, and extra-cellular vesicles (e.g., exosomes), have been employed for the delivery of therapeutic mRNAs in both in vivo and in vitro settings [148,149]. Codiak BioSciences, for instance, to treat solid tumors, has commenced trials of existing therapy based on engineered exosomes in humans. Exosomal mRNA vaccines stimulated more potent IgG and secretive IgA in mice than LNP carriers [150]. Exosomes will be considered a promising mRNA delivery platform due to their lower immunogenicity and toxicity compared to the existing platforms. Thus, they are ideal for multiple mRNA dosing in clinical trials. 

To expand the applications of mRNA therapies, organ- or cell-specific mRNA delivery has been created [150]. Organ-selectively targeting nanoparticle (OSTN) platforms have been developed to deliver mRNAs selectively to mouse lungs, spleen, or liver [151]. The efficiency of mRNA delivery has been significantly increased by a new version of the OSTN platform containing an ionizable phospholipid membrane destabilizing. Alternative administration methods include delivery of mRNA nanoparticles intra-vesical to target bladder sites, orally via robotic pills to target digestive tract sites, and via inhaling to target lung sites [152]. The development of optimized formulations of LNPs or polymer nanoparticles is aimed at enabling the delivery of mRNA specifically to a particular cell type and using cell-specific ligands. In these delivery systems of organ- or cell-targeting mRNA, as more research is conducted, the range of diseases that can be treated with mRNA therapies will broaden [153].

#### 6.1.3. Route of Administration and Storage Considerations of mRNA 

Finding a different route of administration for mRNA can open the doors for newer applications. Using a hyperbranched poly(beta-amino esters) (hPBAEs) nanoformulation, inhalable delivery of mRNA to the lungs has been demonstrated, resulting in a high expression level of the luciferase protein in the lungs of mice. Intranasal administration of vaccines can trigger mucosal immune responses against respiratory infections, making it a promising method for administering COVID-19 vaccines. Oral administration is an excellent alternative route of vaccine delivery owing to its non-invasive nature, patient-care characteristics, and potential for quick implementation. BioNTech, in partnership with Matinas BioPharma, recently developed a potentially novel method for administering oral mRNA vaccines utilizing a lipid nanocrystal platform [154]. Calcium and anionic phospholipids interact to form this lipid nanocrystal, allowing mRNA-inducing active drug molecules to be loaded into the layers [155].

Thermo-stability of mRNA plays a crucial role in developing mRNA into newer applications. The candidate vaccine CVnCoV developed by CureVac can maintain three-month stability at 5 °C and 24 h at ambient temperature; unfortunately, CVn.CoV demonstrated 47% efficacy against COVID-19 in phase 2b/3 of trials [113]. Recently, CureVac has optimized the translation and immunogenicity of CV2. CoV, a second-generation candidate, compared to CVn.CoV. The in vitro study of CV2.CoV exhibited greater protein expression than CVn.CoV-induced high levels of cross-neutralizing antibodies in rats against the SARS-CoV-2 variants (B.1.1.7, B.1.1.298, and B.1.351) [156].

### 6.2. Novel and Versatile Future Therapy of the mRNA Vaccines

#### 6.2.1. The mRNA Vaccines for Infectious Diseases Prevention

The mRNA vaccines have been used in numerous medical fields, including infectious diseases, cancer, genetic disorders, and regenerative medicine [154]. The mRNA vaccines have demonstrated remarkable efficacy and safety against COVID-19 and are being developed for other pathogens, including influenza, Zika, rabies, HIV, and tuberculosis. By delivering mRNA encoding specific antigens to the recipient’s cells, which then produce the antigens and present them to the immune system, mRNA vaccines can induce protective immunity. They can also be designed to target multiple antigens or strains of a pathogen, which could enhance their efficacy and resistance to emerging variants and co-infections [43,155].

#### 6.2.2. The mRNA Vaccines for Cancer Therapy

The mRNA therapy can deliver tumor antigens, immune modulators, or gene-editing components to stimulate antitumor immune responses or correct genetic defects in cancer patients [154]. Tumor antigens are proteins expressed by tumor cells but not normal cells, which the immune system recognizes as foreign. Immune modulators, such as cytokines, checkpoint inhibitors, and costimulatory molecules, can stimulate or inhibit the immune response. Gene editing components include Cas9 (CRISPR), an effector of transcription activator-like nucleases (ETALN), and zinc-finger (ZF) nucleases, which can modify the targeted cell’s genomes. The mRNA therapy can induce tumor cell death, activate immune cells, or modify tumor suppressor genes or oncogenes by delivering mRNA encoding of these molecules to tumor cells or immune cells [154].

#### 6.2.3. The mRNA Vaccines for Genetic Disorders Therapy

The mRNA therapy can replace or supplement missing or faulty proteins or genes in genetic disorders such as hemophilia, cystic fibrosis, and muscular dystrophy [154]. Hemophilia is a bleeding disorder caused by mutations in the clotting factor-encoding genes. Any mutation in the gene that encodes cystic fibrosis transmembrane conductance regulator (CFTR) causes lung disease, i.e., cystic fibrosis. Muscular dystrophy is a disease of the muscles caused by mutations in the dystrophin-encoding gene. By transporting mRNA encoding these proteins or genes to diseased cells or tissues, mRNA therapy can restore their normal function and alleviate disease symptoms [154].

#### 6.2.4. The mRNA Vaccines in Regenerative Medicine 

The mRNA therapy can induce pluripotency, differentiation, or reprogramming of cells for tissue repair or organ transplantation in the regenerative medicine [154]. Pluripotency is a cell’s capacity to differentiate into any body cell. A cell becoming specialized for a particular function is called differentiation. Reprogramming transforms a cell’s identity from one type to another. The mRNA therapy can generate pluripotent stem cells (PSCs), differentiated cells (cardiomyocytes or neurons), or reprogrammed cells (induced PSCs or induced neurons) from somatic cells (fibroblasts or blood cells) by delivering mRNA encoding factors that can influence these processes, such as transcription factors, growth factors, or signaling molecules [154].

### 6.3. Advantages and Acceptance Criteria of mRNA Vaccines

The mRNA vaccines have several advantages over conventional vaccines, which rely on weakened or killed viruses or their components to induce immunity. Some of these benefits include the following [157]:

#### 6.3.1. Speed of Production

Once the genetic sequence of the target antigen is known, mRNA vaccines can be designed and synthesized rapidly. This enables a quicker response to emerging infectious diseases, such as COVID-19;

#### 6.3.2. Scalability 

The mRNA vaccines can be produced in larger quantities using cost-effective and standardized techniques, such as cell-free transcription and purification;

#### 6.3.3. Safety 

The mRNA vaccines do not contain infectious agents or their components, so they are incapable of causing the disease they are designed to prevent. In addition, they do not interfere with cellular functions or integrate into the host genome;

#### 6.3.4. Versatility

The mRNA vaccines can be modified to express different antigens or combinations of antigens, enabling the development of multivalent or universal vaccines against various pathogens. Additionally, they can be altered to improve their stability, delivery, immunogenicity, and specificity;

#### 6.3.5. Effectiveness

In clinical trials, mRNA vaccines have demonstrated high levels of protection against COVID-19, with efficacy rates ranging from 94% to 95%. In addition, they induce both the humoral and cellular immune responses, which are crucial for long-term immunity and protection against emerged variants.

### 6.4. Disadvantages and Challenges of mRNA Vaccines

The future design of safer and more effective mRNA vaccines will unquestionably be aided by a deeper understanding of the limitations of mRNA vaccines, which include the following.

#### 6.4.1. The mRNA Stability 

Enzymes and temperature fluctuations can degrade mRNA molecules, limiting their shelf-life and storage conditions. The mRNA vaccines necessitate ultra-cold storage and specialized transport equipment, which poses logistical challenges and raises costs [158]. Improving the thermal stability of mRNA molecules and developing formulations that can withstand higher temperatures are crucial for future mRNA applications [159].

#### 6.4.2. The mRNA Delivery

The mRNA molecules should be shielded from enzymatic degradation and be efficiently delivered to the target cells. Encapsulating them in protective carriers, such as LNPs, polymeric nanoparticles, or lipid–polymer hybrid nanoparticles, could protect the mRNA from degradation, enhance its uptake and release by cells, and modulate its immunogenicity [158]. LNPs may also cause adverse reactions, such as inflammation, allergic reactions, and immune responses [160].

#### 6.4.3. Accessibility and Acceptance

Accessibility and acceptance of the manufacturing steps of mRNA is another essential factor to be considered. The mRNA vaccines require standardized and high-quality raw materials, such as nucleotides, enzymes, and lipids, and specialized equipment and facilities for their synthesis, purification, and formulation [155]. These requirements create technical and logistical obstacles to the mass production and distribution of mRNA therapy [155]. Some population segments may resist mRNA vaccines due to misinformation, mistrust, and ethical concerns.

#### 6.4.4. Variability

The mRNA vaccines are designed to target a specific antigen, such as the spike protein of SARS-CoV-2. Still, since RNA viruses typically have high mutation rates, they are subjected to evolution over time [161]. This mutation may affect the properties of the virus, such as its transmission and the severity of illness [162]. Moreover, variants might contain mutations that modify the interactions with crucial host components [163]. Notably, selective pressure can result in mutations that allow variants to evade adaptive immune responses [164].

#### 6.4.5. Immunogenicity

Emerging data have demonstrated the importance of humoral and cellular immunity in protecting against viral infections [165]. The mRNA vaccines can induce the innate immune system to regulate antigen-specific immune responses. However, there is a lack of information regarding practical vaccine measurements and descriptions of the duration of the protection provided by mRNA vaccines. Experience with other human coronaviruses indicates that reinfection is possible if the immune response is diminished, and there are still insufficient animal models that can accurately simulate human immune responses [166]. Researchers must continue to search for clinical evidence of the efficacy of mRNA vaccines in human trials [167].

#### 6.4.6. Variation among the Populations 

The immune system is linked to innate and adaptive immunity, and the monitoring of protective immunity in vaccinated populations is an essential factor in determining the efficacy of vaccines. Physical conditions, sex, age, and genetic variations could also affect the vaccine’s effectiveness. The limited availability of testing in many countries has made it difficult to quantify the real risk of infection among individuals [168]. Recent evidence from the World Health Organization (WHO) indicates that the SARS-CoV-2 virus, for instance, can spread primarily among young people, and those under 40 are more susceptible to infection [169]. The mRNA-based antiviral vaccines have distinct advantages over conventional vaccines, but there is still debate regarding the effect of immunization on various populations [9,170]. Additional research must be conducted to comprehend the vaccination status of multiple individuals. Understanding the mechanisms underlying individual differences in the efficacy of mRNA vaccines will aid in advancing mRNA vaccines.

#### 6.4.7. Adverse Reactions

As with any vaccine, mild to moderate adverse reactions are possible after vaccination [171]. Some severe side effects may occur, leading to fear and anxiety among the population [172]. Theoretically, the mRNA vaccine is safer than other types of vaccines, as there is no infectious virus involved in its production, and there is a reduced risk of virulence reversion and insertional mutagenesis [173]. Due to the small number of participants, clinical trials may be inherently incapable of assessing the safety of vaccines, so comprehensive and long-term vaccine safety monitoring is required. By analyzing the data, not all immediate reactions to vaccinations are genuine allergic reactions [174]. Vaccine components, such as excipients, inactive ingredients, and liposomal delivery vehicles, are associated with allergic reactions [110]. Studies have shown that the incidence of adverse reactions is correlated with dosage, but whether dosage affects the occurrence of adverse reactions remains to be investigated [175,176].

Moreover, individual differences in the innate immune system should not be ignored [177]. Evaluations of the safety profile of mRNA vaccines are currently limited by sample size, comparator group, follow-up time, age, gender, and racial/ethnic group of vaccinated individuals. Further investigation of the associations between vaccination and associated adverse events will result in a more thorough evaluation of the safety of mRNA vaccine candidates. Finally, it is crucial to continue public education and communication, which are necessary to increase awareness and confidence in mRNA vaccines’ safety and efficacy, as they can potentially improve global health and save millions of lives.

## 7. Conclusions

This review paper highlighted the mRNA vaccines’ significant role in revolutionizing infectious disease prevention and therapeutic interventions. Traditional vaccine platforms have limitations in scalability, speed of development, and adaptability to emerging pathogens, while mRNA vaccines offer a versatile and rapid platform for targeted vaccine development. The modular nature of mRNA vaccines allows for easy adaptation to different pathogens, and their rapid production capabilities provide flexibility in responding to any infectious diseases that could emerge. However, the efficient delivery of mRNA molecules poses a challenge, which was addressed by using nanoparticles as delivery systems. LNPs have emerged as the leading class of nanoparticles for mRNA vaccine delivery, providing protection, efficient cellular uptake, and controlled release of the mRNA payloads. 

Key points discussed in this review include the different types of mRNA vaccines, their mechanism of action, and the optimization processes involved in their development. Clinical trials of mRNA vaccines, particularly in the context of COVID-19, have shown promising results in terms of safety and efficacy. Lessons learned from these trials will guide future research and development efforts. The future applications of mRNA vaccines extend beyond infectious disease prevention. They hold potential in various healthcare areas, including cancer immunotherapy and addressing diseases with a genetic component. The advantages and limitations of mRNA vaccines for different diseases were outlined, along with the challenges that should be overcome.

In summary, mRNA vaccines offer exciting possibilities for the future of infectious diseases and cancers. Their flexibility, adaptability, and rapid production capabilities position them as valuable tools in disease prevention and therapeutic interventions. Further research is needed to explore their full potential, address challenges, and optimize their use in different disease contexts. Moving forward, it is recommended that future research focuses on improving vaccine design, delivery systems, and manufacturing processes. Understanding the long-term safety and durability of immune responses generated by mRNA vaccines is essential. Additionally, exploring the combination of mRNA vaccines with other immunotherapeutic approaches may further enhance their efficacy. Continued research and collaboration in mRNA vaccines will pave the way for innovative disease prevention and treatment solutions.

## Figures and Tables

**Figure 1 pharmaceutics-15-01972-f001:**
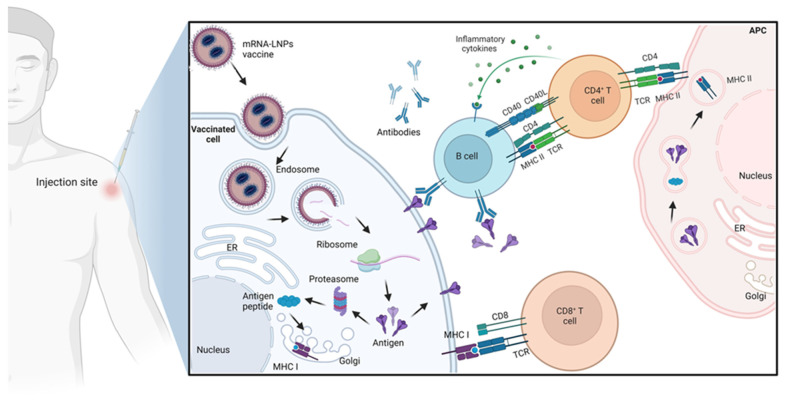
Mechanism of immune activation by mRNA-LNPs vaccines: Upon administration, mRNA-LNPs are internalized by immune cells, where the mRNA is released from the LNPs. The mRNA is then recognized by ribosomes, leading to the translation of antigenic proteins. These proteins are processed and degraded by proteasomes producing small peptides that then get loaded on major histocompatibility complex class I (MHC I) molecules and presented on the cell surface, which binds to the T cell receptor (TCR), activating CD8+ T cells to kill infected cells. The produced antigen can also be endocytosed by antigen-presenting cells (APCs) and further degraded by the lysosomes producing small peptides that then get loaded on major histocompatibility complex class II (MHC II) molecules for recognition by CD4+ T cells, which subsequently activate both the cellular immune responses by secreting inflammatory cytokines and the humoral immune responses by activating B cells.

**Figure 2 pharmaceutics-15-01972-f002:**
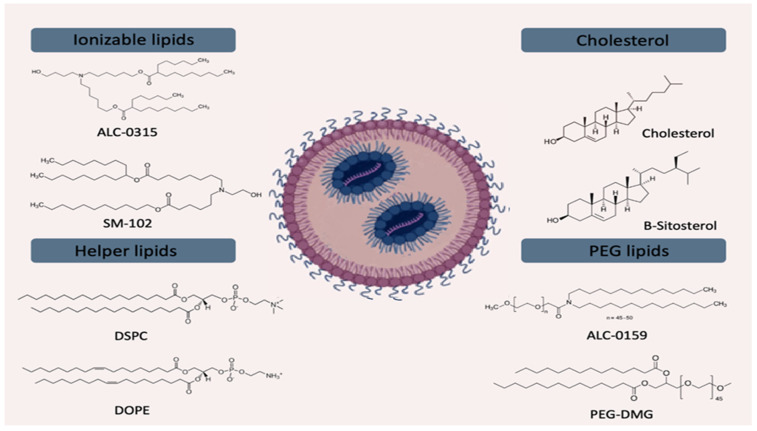
The main composition of LNPs for nucleic acid delivery. The LNPs showed mRNA encapsulated into the core of the nanocarrier. ALC-0315, [(4-Hydroxybutyl)azanediyl]di(hexane-6,1-diyl) bis(2-hexyldecanoate); SM-102, 8-[(2-hydroxyethyl)[6-oxo-6-(undecyloxy)hexyl]amino]-octanoic acid, 1-octylnonyl ester; DSPC, 1,2-distearoyl-sn-glycero-3phosphocholine; DOPE, 1,2-dioleoyl-sn-glycero-3-phosphoethanolamine; ALC-0159, Methoxypolyethyleneglycoloxy(2000)-N,N-ditetradecylacetamide; PEG-DMG, 1,2-Dimyristoyl-rac-glycero-3-methoxypolyethylene glycol.

## Data Availability

The authors confirm that the data supporting the findings of this study are available within the article.

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
