# Peer review of "Recent Advancement in mRNA Vaccine Development and Applications"

_pharmaceutics, 2023, doi:10.3390/pharmaceutics15071972_

Round 1
Reviewer 1 Report
This paper presents the recent advancement in mRNA vaccine development and applications. The different types of mRNA vaccines, their modes of action, and the process of optimizing and developing them and overcoming their limitations are reviewed, and the definition, mechanism of action, delivery systems, manufacturing processes, and clinical applications of mRNA vaccines are discussed. In addition. For an introduction to the various delivery systems, the role of nanoparticles in mRNA vaccine delivery is outlined with emphasis.
Main issues:
1. The study of the mechanism of action of mRNA vaccines is mentioned in the paper, but the content is simple and does not elaborate on the mechanism of action. The understanding of the mechanism of action of mRNA vaccines can provide a strong support for the design of mRNA vaccines.
2. mRNA optimization section: How many pathways can be used to enhance the stability and efficiency of mRNA vaccines? Do these pathways enhance the safety or the protective efficiency of mRNA vaccines? Or are both enhanced? The presentation here lacks organization.
3. The abstract mentions that the article focuses on outlining the role of nanoparticles in mRNA vaccine delivery, but the body of the article is scattered in sorting out its content.
Secondary issues:
1. some mRNA vaccines can be put into clinical use after verifying their good safety and protection efficiency through preclinical and clinical trials. However, the logical order of these two parts can be adjusted, since the article firstly reviews the clinical application of mRNA vaccines and then introduces the content of clinical trials.
2. Some subheadings of the article do not accurately summarize the content.
3. The content of the abstract is incomplete. The disadvantages and prospects of mRNA vaccines are not mentioned.
Minor editing of English language required
Author Response
Main issues:
- The study of the mechanism of action of mRNA vaccines is mentioned in the paper, but the content is simple and does not elaborate on the mechanism of action. The understanding of the mechanism of action of mRNA vaccines can provide a strong support for the design of mRNA vaccines.
The mechanism of action of mRNA in Section 2.3 was amended accordingly. Additions are yellow-highlighted.
- mRNA optimization section: How many pathways can be used to enhance the stability and efficiency of mRNA vaccines? Do these pathways enhance the safety or the protective efficiency of mRNA vaccines? Or are both enhanced? The presentation here lacks organization.
Thank you for your comment. We agree that the flow was not correct so we switched sections 2.2 (Optimizing mRNA vaccines) and 2.3 (Mechanism of action of mRNA vaccines). We also amended the section (Optimizing mRNA Vaccines) accordingly. Additions are yellow-highlighted.
- The abstract mentions that the article focuses on outlining the role of nanoparticles in mRNA vaccine delivery, but the body of the article is scattered in sorting out its content.
We highlighted the use of 4 classes of delivery systems in section 3. All could be considered nano-delivery systems, particularly nanoparticles.
Secondary issues:
- Some mRNA vaccines can be put into clinical use after verifying their good safety and protection efficiency through preclinical and clinical trials. However, the logical order of these two parts can be adjusted, since the article firstly reviews the clinical application of mRNA vaccines and then introduces the content of clinical trials.
Most studies on mRNA testing its application against infectious diseases or cancers. They are the two main areas that mRNA is currently being applied for. That is why we highlighted them in section 4 and then we demonstrated the mRNA vaccines that are currently in clinical trials. As shown in Table 1, all mRNA vaccines are applied for infectious diseases of cancers as well. Later, we highlighted the Novel and Versatile Future Therapy of mRNA Vaccines, which there are other areas of interest such as Genetic Disorders (6.2.3.) and Regenerative Medicine (6.2.4.).
- Some subheadings of the article do not accurately summarize the content.
Thank you for this comment. We reviewed all subheadings and adjust some. The amended subheadings are yellow-highlighted.
- The content of the abstract is incomplete. The disadvantages and prospects of mRNA vaccines are not mentioned.
Thank you for this comment. We amended the abstract accordingly. Additions are yellow-highlighted.
Reviewer 2 Report
This manuscript is a review on mRNA vaccines. Apart from a few typos and some grammatical/stylistic errors, the manuscript is globally well-written and provides a sound overview of this technology.
There are however a few statements that merit amendment.
Page 2, the sentence “trigger the immune responses against synthetic mRNA molecules encoding specific viral or bacterial antigens” is erroneous. The authors mention just before that mRNA will use cell machinery to produce antigens, which is precisely the case, but the immune responses will be directed to these antigens, not to the mRNA.
The mention of “nuclear targeting” is also erroneous, as mRNA will be processed by ribosomes in the cytoplasm as well mentioned later.
The sentence “degraded by proteasomes producing major histocompatibility complex class I (MHC I)” in the legend of Figure 1 is not correct and should read “degraded by proteasomes producing small peptides that then get loaded on major histocompatibility complex class I (MHC I) molecules”. Similarly for Class II molecules, the proper statement is :”degraded by the lysosomes producing small peptides that then get loaded on major histocompatibility complex class II (MHC II) molecules”.
Figure 2 show different types of nucleotides in the particle, this should be explained in the legend or the figure could be changed to show only mRNA in the LNP.
The paragraph on protamine is obscure and needs rewriting.
The mechanism of dengue-reactivating antibodies after vaccination to Zika should be better explained.
The reason for adding pembrolizumab should be explained as readers may not be aware of immune checkpoints. The same goes for the paragraph about bispecific antibodies.
The mention of “mRNA antigens” in paragraph 5 after table 1 is unclear. In the same paragraph “One of the most aspects of vaccine safety” is clearly an incomplete statement.
“hPBAE” should be explained.
Finally, there are numerous redundancies indicating that the manuscript could be shortened and made more to the point while also providing better explanations where shortcuts have been noticed.
As mentioned the menuscript could be shortened/reorganized but must anyway benefit from a thorough re-reading.
Author Response
- Page 2, the sentence “trigger the immune responses against synthetic mRNA molecules encoding specific viral or bacterial antigens” is erroneous. The authors mention just before that mRNA will use cell machinery to produce antigens, which is precisely the case, but the immune responses will be directed to these antigens, not to the mRNA.
Thank you for this comment. The sentence has been amended and yellow highlighted accordingly.
- The mention of “nuclear targeting” is also erroneous, as mRNA will be processed by ribosomes in the cytoplasm as well mentioned later.
The sentence has been corrected as per suggestion.
- The sentence “degraded by proteasomes producing major histocompatibility complex class I (MHC I)” in the legend of Figure 1 is not correct and should read “degraded by proteasomes producing small peptides that then get loaded on major histocompatibility complex class I (MHC I) molecules”. Similarly for Class II molecules, the proper statement is :”degraded by the lysosomes producing small peptides that then get loaded on major histocompatibility complex class II (MHC II) molecules”.
Thank you for pointing out this comment. The legend has been amended accordingly.
- Figure 2 show different types of nucleotides in the particle, this should be explained in the legend or the figure could be changed to show only mRNA in the LNP.
The figure has been changed to show only mRNA in the LNPs as per suggestion.
- The paragraph on protamine is obscure and needs rewriting.
The protamine section has been modified significantly.
- The mechanism of dengue-reactivating antibodies after vaccination to Zika should be better explained.
The mechanism of dengue-reactivating antibodies has been expanded as per suggestion. This paragraph has been added:
(Encapsulating modified mRNA encoding Zika virus structural genes into lipid-based nanocarriers is a versatile platform to produce mRNA-based Zika vaccine. The administration of a prime-boost vaccine of modified mRNA encoding Zika prM-E genes that generated virus-like particles lead to an increase in the levels of nAbs that protected immunocompromised and immunocompetent mice. Hence, the administrated mRNA Zika vaccine-induced protective antibody responses against Zika virus).
- The reason for adding pembrolizumab should be explained as readers may not be aware of immune checkpoints. The same goes for the paragraph about bispecific antibodies.
The reason for adding pembrolizumab chemotherapy and administration in combination with mRNA in the treatment of melanoma has been extended. Similarly, the trial of BNT111 with cemiplimab has been expanded.
- The mention of “mRNA antigens” in paragraph 5 after table 1 is unclear. In the same paragraph “One of the most aspects of vaccine safety” is clearly an incomplete statement.
The whole paragraph has been rewritten to be clearer.
- “hPBAE” should be explained.
The explanation of hPBAE (hyperbranched poly(beta-amino esters)) has been added.
- Finally, there are numerous redundancies indicating that the manuscript could be shortened and made more to the point while also providing better explanations where shortcuts have been noticed.
Thank you for this comment. We went through the manuscript and adjusted the order of some sections (e.g. 2.2 and 2.3) to maintain a good flow of the manuscript. We also amended the subheadings to explain better their content. Finally, we proofread the whole manuscript.